# Improved Antioxidative, Anti-Inflammatory, and Antimelanogenic Effects of Fermented Hydroponic Ginseng with *Bacillus* Strains

**DOI:** 10.3390/antiox11101848

**Published:** 2022-09-20

**Authors:** Ji-Young Park, Myung Wook Song, Kee-Tae Kim, Hyun-Dong Paik

**Affiliations:** 1Department of Food Science and Biotechnology of Animal Resources, Konkuk University, Seoul 05029, Korea; 2Research Laboratory, WithBio Inc., Seoul 05029, Korea

**Keywords:** *Panax ginseng*, hydroponic ginseng, antioxidant activity, anti-inflammatory effect, antimelanogenic effect, *Bacillus* fermentation

## Abstract

Compared with traditionally cultured ginseng, hydroponic ginseng (HG) contains more remarkable bioactive compounds, which are known to exert diverse functional effects. This study aimed to enhance the multifunctional effects, including the antioxidative, anti-inflammatory, and antimelanogenic effects, exhibited by fermented HG with *Bacillus* strains, such as *Bacillus subtilis* KU43, *Bacillus subtilis* KU201, *Bacillus polyfermenticus* SCD, and *Bacillus polyfermenticus* KU3, at 37 °C for 48 h. After fermentation by *B. subtilis* KU201, the antioxidant activity, determined using ABTS and FRAP assays, increased from 25.30% to 51.34% and from 132.10% to 236.27%, respectively, accompanied by the enhancement of the phenolic compounds and flavonoids. The inflammation induced in RAW 264.7 cells by lipopolysaccharide (LPS) was ameliorated with fermented HG, which regulated the nitric oxide (NO), prostaglandin E_2_ (PGE_2_), and proinflammatory markers (tumor necrosis factor (TNF)-α, and interleukin (IL)-1β and IL-6). The treatment with fermented HG inhibited the melanin accumulation in B16F10 cells induced by α-melanocyte-stimulating hormone (α-MSH) by controlling the concentrations of melanin synthesis and tyrosinase activity. These results indicate that the HG exhibited stronger antioxidative, anti-inflammatory, and antimelanogenic effects after fermentation. Consequently, HG fermented by *Bacillus* strains can potentially be used as an ingredient in cosmetological and pharmaceutical applications.

## 1. Introduction

*Panax ginseng* Meyer, which has a long history of pharmacological effects, has been consumed as a valuable herb for enhancing the capacity to cope with fatigue and improve physiological conditions [1]. Ginseng comprises various active components, such as saponins, flavonoids, polysaccharides, organic acids, amino acids, sterols, peptides, and minerals [2]. Although the general pharmacological effects of ginseng are complicated due to the different properties and multiple functions of its active constituents, the main biofunctional activities of ginseng include antioxidant, antimelanogenic, anti-inflammatory, immunoregulatory, and neuroprotective effects [3,4].

Both the root and shoot (leaf and stem) parts of the ginseng plant exhibit various bioactive effects [5]. In hydroponically cultivated ginseng that is harvested as a whole body, the shoots possess a higher level of ginsenosides (Rg1, Rg2 + Rh1, Rd, and Rg3) and phenolic compound (*p*-coumaric acid) than the primary roots [6]. This is attributed to the better assimilation of these compounds during the early stages of growth in the shoots than in the roots during the same cultivation period [6]. This results in hydroponic ginseng (HG) exerting a higher antioxidant effect than soil-cultivated ginseng. Moreover, HG cultivation is ecofriendly and cost-effective because it is cultivated without pesticides, and the key growth factors are easier to control in hydroponic cultivation systems [7]. Despite the potential for commercially scaling up the hydroponic method of ginseng cultivation to harness it as a functional ingredient, there are few studies that have investigated the pharmacological effects of HG and its fermented products [5,8].

The tonic and therapeutic effects of ginseng have recently re-established interest in the biological techniques of fermentation [3]. The content and bioavailability of phytochemicals are significantly influenced by the fermentation process. With the aid of microorganisms, the bound forms of phytochemicals in natural plants can be biotransformed to their free forms, thereby improving the bioavailability and bioactivity [9]. Moreover, the absorption rate of phytochemicals is improved by fermentation, owing to the high membrane permeability [10]. Therefore, the use of diverse fermented natural sources of therapeutic materials is increasing due to the benefits, such as the functional properties, nontoxicity, and safety [11].

According to previous research, fermented ginseng that contains the ginsenosides Rg3 and Rh2 diminished the induced reactive oxygen species (ROS) and nitrite levels more strongly than did nonfermented ginseng in lipopolysaccharide (LPS)-stimulated RAW 264.7 cells, implying that fermented ginseng has improved antioxidative and anti-inflammatory activities [12]. *L**actobacillus plantarum* KP-4 fermentation promoted minor ginsenoside content and alleviated LPS-induced alanine transaminase, aspartate transaminase, and inflammation-related cytokines (interleukin (IL)-6, IL-1β, and tumor necrosis factor (TNF)-α) through the TLR4/MAPK pathway in mice [13]. In addition, fermented ginseng showed significantly improved bioavailability compared with nonfermented ginseng as a cosmeceutical ingredient, with the advantages of higher skin-permeation, intestinal-permeability, and ginsenoside levels [14].

Oxidative stress is associated with inflammation and skin pigmentation, and it ultimately induces inflammatory responses and melanogenesis [15]. Therefore, in this study, we investigated fermented HG as a cosmetical and functional ingredient due to its multiple functional effects, including its antioxidative, anti-inflammatory, and antimelanogenic effects.

## 2. Materials and Methods

### 2.1. Materials

HG was purchased from Ginseng Well-Life (Gwangju, Korea). Chemical reagents, including 3-(4,5-Dimethylthiazol-2-yl)-2,5-diphenyltetrazolium bromide (MTT) reagent, α-melanocyte-stimulating hormone (α-MSH), and L-3,4-dihydroxyphenylalanine (L- DOPA), were purchased from Sigma-Aldrich (St. Louis, MO, USA).

### 2.2. Bacteria Strain and Cell Culture Conditions

*Bacillus subtilis* KU43, *Bacillus subtilis* KU201, *Bacillus polyfermenticus* SCD, and *Bacillus polyfermenticus* KU3, isolated from kimchi, were used for fermentation. These *Bacillus* strains were cultured in TSB (BD Biosciences; NJ, USA) in a shaking incubator at 37 °C for 24 h (150 rpm). Murine RAW 264.7 macrophage cell line and murine B16F10 melanoma cell line were obtained from Korean Cell Line Bank (KCLB; Seoul, Korea), and were incubated in Dulbecco’s modified Eagle’s medium (DMEM) (Hyclone; Logan, UT, USA) supplemented with 10% fetal bovine serum (FBS) (Hyclone; Logan, UT, USA) and 1% penicillin/streptomycin (P/S) (Hyclone; Logan, UT, USA) at 37 °C under 5% CO_2_ condition.

### 2.3. Preparation of HG

Washed HG was prepared for fermentation. After HG was pulverized into powder, 50 g of powder was refluxed with 500 mL of 50% ethanol at 65 °C for 6 h. This process was repeated thrice with the remaining residue. Whatman No. 2 filter paper was used for filtering ethanol extract; this filtered extract was condensed at 60 °C with a N-1000 V rotary evaporator (EYELA; Tokyo, Japan). The dark ginseng syrup was diluted with distilled water and stored at −20 °C [16].

### 2.4. Liquid-State Fermentation of HG

HG was prepared and fermented with little modification of the previously described method [17]. HG (10 mg/mL) was inoculated with 1% *Bacillus* strain culture (1 × 10^8^ CFU/mL) after sterilization at 90 °C for 10 min. Fermentation proceeded at 37 °C for 48 h. The fermented HG was centrifuged at 8000× *g* at 4 °C for 15 min, and the collected supernatant was filtrated through a 0.45 μm membrane (Advantec; Tokyo, Japan).

### 2.5. Total Phenolic and Total Flavonoid Contents

As per Hwang et al. [8], the phenolic and flavonoid contents were measured. The concentration of phenolics was assayed by initially mixing the sample (50 μL) and 2% NaCO_3_ (1 mL), and incubating for 5 min. Then, 50% Folin–Ciocalteu phenol reagent (50 μL) was added.

The concentration of flavonoids was measured by initially mixing the sample (100 μL), 5% NaNO_2_ (20 μL), and 60% ethanol (800 μL). After a 6 min reaction, 10% aluminum chloride (20 μL) was added. After another 6 min reaction, 4% sodium hydroxide (60 μL) was added. Each mixed solution of TPC and TFC assays were reacted for 30 min in the final step, and absorbance was measured at 750 nm and 405 nm, respectively. Phenolic content was expressed as mg/g gallic acid equivalents (GAEs), and flavonoid content as mg/g quercetin equivalents (QEs) [18].

### 2.6. 2,2′-Azino-bis(3-ethylbenz-thiazoline-6-sulfonic Acid (ABTS) and Ferric Reducing Antioxidant Power (FRAP) Assay

To prepare ABTS reagent, 14 mM ABTS and 5 mM potassium persulfate each were dissolved in 0.1 M potassium phosphate buffer (pH 7.4) and mixed in equal volume. After 16 h of incubation, the reagent was diluted to an absorbance of 0.7 at 734 nm. The sample (50 μL) and ABTS reagent (950 μL) were incubated for 15 min. Absorbance was measured at 734 nm [16]. 

For the FRAP assay, the FRAP reagent was produced by warming 10 mM of 2, 4, 6-tri [2-pyridyl]-s-triazine (TPTZ) in 40 mM HCl for 30 min at 50 °C. After preheating, 10 mM TPTZ, 20 mM ferric chloride, and 300 mM acetate buffer (pH 3.6) were mixed in a 1:1:10 ratio and incubated for 15 min at 37 °C. The sample (50 μL) and mixed reagents (950 μL) were allowed to react for 30 min. The absorbance was measured at a wavelength of 593 nm. The antioxidant rate was calculated as ferrous sulfate equivalents using the FeSO_4_ standard curve [16].

### 2.7. Cell Viability of RAW 264.7 and B16F10 Cells

The anti-inflammatory effects of HG were measured using RAW 264.7 cells [19]. To evaluate cell viability, RAW 264.7 cells (1 × 10^5^ cells/well) were plated and incubated for 24 h. Samples and DMEM were added to the seeded cells. For the control group, the sample was replaced with DMEM. After 24 h of incubation, the cells were reacted with 0.5 mg/mL MTT reagent for 1 h to produce purple formazan. The formazan, which remained after discarding the MTT reagent, was dissolved in dimethyl sulfoxide (DMSO) and measured at 570 nm.

B16F10 cells were used to assay the antimelanogenic effects of the samples [20]. The cytotoxicity of the sample in B16F10 cells was measured with the MTT assay. B16F10 cells (5 × 10^3^ cells/well) were cultured for 24 h. The cells were also incubated in combinations of different concentrations of the samples and DMEM for 48 h. The samples were replaced with DMEM in the control group. The crystal formazan, which was produced by reacting cells with 0.5 mg/mL MTT agent for 2 h, was dissolved in DMSO, and the absorbance was measured at 570 nm.

### 2.8. Cellular Nitric Oxide (NO) Production in RAW 264.7 Cells

A nitric oxide (NO) assay was performed by adding samples and 1 μg/mL of LPS to the seeded cells that were incubated for 2 h. Both the samples and LPS were replaced with DMEM in the negative control. The sample was replaced with DMEM in the positive control. After 24 h, the collected supernatant and Griess reagent were reacted in a 1:1 ratio for 15 min. Absorbance was measured at a wavelength of 540 nm [19]. The produced NO was represented in sodium nitrite equivalents using a NaNO_2_ standard curve.

### 2.9. Cellular Tyrosinase Activity and Melanin Content in B16F10 Cells

After preincubating in 2 mL of media, the B16F10 cells (1 × 10^5^ cells/well) were plated in 6-well plates for 24 h. The cells were cotreated with samples and 0.2 μg/mL of α-MSH for 48 h. Cells were lysed with a mixture of 0.01 mol/L sodium phosphate buffer (pH 7.0) and 0.1 mmol/L phenylmethanesulfonylfluoride fluoride (PMSF). The lysate was centrifuged to obtain the supernatant, which was reacted with 10 mM L-DOPA for 1 h at 37 °C. Absorbance was measured at 540 nm [21].

To estimate the melanin production rate, B16F10 cells treated with the sample and α-MSH were detached using trypsin treatment followed by centrifugation. The supernatant was discarded, and the sediments were resolved with 1N NaOH for 1 h at 80 °C. Absorbance was measured at 405 nm [22].

### 2.10. L-DOPA Staining Assay of B16F10 Cells

B16F10 cells (1 × 10^5^ cells/well) were placed in a 6-well plate for 24 h and incubated with the sample and α-MSH for 48 h. B16F10 cells were fixed with 10% formaldehyde for 20 min; these fixed cells were washed twice again, and reacted with 10 mmol/L L-DOPA at 37 °C for 3 h [21]. Pigmentation of B16F10 cells was observed with an Eclipse Ti2-U microscope (Nikon Co.; Tokyo, Japan) attached to a DS-Ri2 digital camera (Nikon Co.; Tokyo, Japan).

### 2.11. Relative Quantification of Gene Expression by Quantitative Real-Time PCR

Relative gene expression levels were quantified by qRT-PCR following a previously described procedure of cell culture. Cellular RNA was extracted using the RNeasy Mini kit (Qiagen; Hilden, Germany), and cDNA was synthesized with the SensiFAST cDNA synthesis kit (Bioline; London, UK), based on the RNA quantification result. Relative gene expression levels were determined using QuantStudio 1 real-time PCR (Thermo Fisher Scientific; Waltham, MA, USA) with the SensiFAST SYBR NO-ROX mix (Bioline; London, UK). Real-time PCR proceeded according to Kim et al. [18] and Chatatikun et al. [23]. The relative gene expression levels were analyzed as 2^−ΔΔCq^ values for each sample relative to *β-Actin*. All the specific primers are shown in Table 1 (Bioneer Co.; Daejeon, Korea) [20,24].

### 2.12. Enzyme-Linked Immunosorbent Assay (ELISA)

Cell-free supernatant was taken from RAW 264.7-cell medium treated with samples and LPS, which is the same condition as the NO assay, and it was diluted according to the manufacturer’s instructions. The inhibitory activity of the sample against TNF-α, IL-1β, and IL-6 (Invitrogen; Vienna, Austria), which are proinflammatory cytokines, and prostaglandin E_2_ (PGE_2_) (R&D systems; Minneapolis, MN, USA), were measured using an enzyme-linked immunosorbent assay (ELISA) kit, according to a previously described method [19].

### 2.13. Statistical Analysis

Each result is represented as the mean ± standard deviation of triplicate experiments. All results were assessed using one-way analysis of variance (ANOVA) with Duncan’s multiple range test to compare significance, and they were considered statistically significant at p < 0.05, using IBM SPSS software (version 18.0, SPSS Inc.; Chicago, IL, USA).

## 3. Results

### 3.1. Growth of Bacillus Stains during Fermentation of HG

The viable cell numbers of the *Bacillus* strains in the HG changed after 48 h of fermentation (Figure 1). All strains (*B. subtilis* KU43, *B. subtilis* KU201, *B. polyfermenticus* SCD, and *B. polyfermenticus* KU3) were inoculated at a final concentration of 6 log CFU/mL. The total number of *B. subtilis* KU201 and *B. polyfermenticus* SCD in the HG increased to 8.02 and 7.98 log CFU/mL, respectively, during 48 h of fermentation. *B. subtilis* KU43 also increased to 7.63 log CFU/mL over 16 h, and it decreased to 7.07 log CFU/mL. Similarly, *B. polyfermenticus* KU3 increased to 8.01 log CFU/mL and decreased to 7.00 log CFU/mL. The growth of microbial strains can be attributed to phytochemical bioconversion, including the polyphenolic compounds and ginsenosides in HG.

### 3.2. Total Phenolic Content (TPC), Total Flavonoid Content (TFC), and Antioxidant Activities (ABTS and FRAP)

The TPC and TFC of the nonfermented and fermented HG were compared (Table 2). The phenolic compound content, which showed a high rate in hydroponic ginseng in a prior study [8], was significantly higher after fermentation. The total flavonoid content also increased with fermentation by *Bacillus* strains. Especially, 51.62 mg GAE/g of the phenolic compounds in the nonfermented HG gradually increased after the fermentation by *B. subtilis* KU201 to 90.22 mg GAE/g, and 24.33 mg QE/g of the flavonoids in the nonfermented HG also increased to 43.27 mg QE/g. In addition, the antioxidant activities (ABTS and FRAP) were measured, as shown in Table 2. The results of the ABTS assay showed 25.30% radical-scavenging activity by the nonfermented HG, whereas those of the fermented HG were 34.00, 51.34, 36.88, and 40.62% with *B. subtilis* KU43, *B. subtilis* KU201, *B. polyfermenticus* SCD, and *B. polyfermenticus* KU3, respectively. Similarly, the radical-scavenging activity in the FRAP assay increased after fermentation with *B. subtilis* KU43, *B. subtilis* KU201, *B. polyfermenticus* SCD, and *B. polyfermenticus* KU3 from 132.1 μM to 176.30, 236.27, 207.86, and 199.66 μM, respectively. This increase in the antioxidant activities in the HG could have been caused by the phenolic compound and flavonoid contents, which indicated similar results.

### 3.3. Anti-Inflammatory Effect in LPS-Induced RAW 264.7 Cells

#### 3.3.1. Cell Viability and Cellular NO Production

Prior to investigating the anti-inflammatory effects of the nonfermented and fermented HG, the maximum nontoxic dosage of RAW cells was identified with the cell viability. As shown in Figure 2a, the nontoxic concentration with the maximum cell viability was assessed as 1 mg/mL HG, and further experiments verifying the anti-inflammatory effects were conducted with this concentration.

As shown in Figure 2b, the LPS treatment without HG induced the production of 40.98 μM of NO. Although the treatment with 1 mg/mL of nonfermented HG reduced the LPS-induced NO to 21.41 μM, the treatment with *B**. subtilis* KU43-, *B**. subtilis* KU201-, *B**. polyfermenticus* SCD-, and *B**. polyfermenticus* KU3-fermented HG decreased the rates to 19.61, 6.08, 15.78, and 8.35 μM, respectively. Consequentially, the NO production was diminished more significantly with the treatment of fermented HG than with that of the nonfermented HG, and it showed a significant inhibition rate with the treatment of *B**. subtilis* KU201-fermented HG.

#### 3.3.2. mRNA Expressions of *iNOS, COX-2*, *TNF-α*, *IL-1β*, and *IL-6*

Because high inhibition rates of LPS-induced NO were shown by the *B**. subtilis* KU201-, *B**. polyfermenticus* SCD-, and *B**. polyfermenticus* KU3-fermented HG among the fermented HG, the mRNA expressions of the inflammatory mediators were identified with these samples. As shown in Figure 3, all the proinflammatory enzymes and cytokines showed similar tendencies. The mRNA expressions of *inducible*
*nitric oxide synthase (iNOS), cyclooxygenase-2 (COX-2), TNF-α, IL-1β,* and *IL-6*, which increased with the LPS stimulation, were more downregulated in the fermented-HG-treated group than in the nonfermented-HG-treated group (Figure 3). Similar to the previous experiment, *B**. subtilis* KU201-fermented HG alleviated the gene expressions of *iNOS* and *COX-2* to 0.14- and 0.13-fold those of the positive control, respectively. Moreover, *B**. subtilis* KU201-fermented HG reduced the gene expressions of *TNF-α, IL-1β,* and *IL-6* to 0.15-, 0.30-, and 0.17-fold those of the positive control, respectively.

#### 3.3.3. Protein Concentrations of PGE_2_ and Cytokines (TNF-α, IL-1β, and IL-6)

As shown in Figure 4a, the inhibitory rate of PGE_2_ was higher with the treated fermented HG than with the treated nonfermented HG. The fermented HG treatment decreased the rate of PGE_2_ production by approximately half that of the positive control. Although the LPS stimulation promoted the expressions of cytokines, the HG dramatically alleviated the levels of cytokines (TNF-α, IL-1β, and IL-6) (Figure 4b–d). In particular, the *B**. subtilis* KU201-fermented HG alleviated the IL-1β and IL-6 levels by a similar rate as that in the negative control group (Figure 4c,d).

### 3.4. Antimelanogenic Effects in B16F10 Cells

#### 3.4.1. Cell Viability, Cellular Melanin Content, and Cellular Tyrosinase Activity

To evaluate the antimelanogenic effects of HG, its cytotoxicity in B16F10 cells was determined. Because an HG concentration of 0.25 mg/mL showed no cytotoxicity, further experiments for assessing the antimelanogenic effects were performed at this concentration (Figure 5a). For the induction of melanogenesis, the α-MSH-stimulated group was used as the positive control. Based on nontoxic concentrations, the intracellular melanin production was inhibited, and the concentrations fell from 136.94% to 106.39%, 100.56%, 105.28%, and 115.565% after the treatment with *B**. subtilis* KU43-, *B**. subtilis* KU201-, *B**. polyfermenticus* SCD-, and *B**. polyfermeticus* KU3-fermented HG, respectively (Figure 5b). Tyrosinase, which is the key enzyme involved in melanin synthesis, was measured as described by Park et al. [21]. As shown in Figure 5c, the cellular tyrosinase activity was promoted to 193.47% by the α-MSH treatment compared with that of the negative control. The activated tyrosinase rate decreased with the nonfermented HG pretreatment, whereas the *B**. subtilis* KU43, *B**. polyfermenticus* SCD, and *B**. polyfermenticus* KU3 fermentation groups showed similar tyrosinase activities of 111.98, 117.29, and 116.41%, respectively; the *B**. subtilis* KU201-fermented HG group indicated the lowest tyrosinase activity (83.70%). These results suggest that fermented HG, and especially after fermentation with *B**. subtilis* KU201, can more highly suppress the cellular tyrosinase activation and lead to an inhibitory effect on melanogenesis in α-MSH-stimulated B16F10 cells than nonfermented HG.

#### 3.4.2. L-DOPA Staining

The L-DOPA staining assay is another method for the microscopic identification of the tyrosinase activity. In Figure 6, the cells of the positive control group with α-MSH showed black spots and dendrite development compared with the cells of the negative control group without α-MSH. Moreover, all the cells from the HG-pretreated groups showed fewer dendrites and dark spots than the cells of the positive group. In particular, the *B**. subtilis* KU43-, *B**. subtilis* KU201-, and *B**. polyfermenticus* SCD-fermented HG more gradually ameliorated the α-MSH-induced spots and dendritic processes than did the nonfermented HG, indicating that fermented HG better inhibits the tyrosinase activity.

#### 3.4.3. Gene Expressions of MITF and Melanogenic Proteins (Tyrosinase, TRP-1, and TRP-2)

To identify the mechanism by which HG inhibits melanogenesis, the gene expression levels of *microphthalmia-associated transcription factor (MITF), tyrosinase, tyrosinase-related protein (TRP)-1,* and TRP-2 were measured (Figure 7). These results demonstrated that the fermentation by the *Bacillus* strains gradually decreased the *MITF, Tyrosinase, TRP-1, and TRP-2* mRNA levels in the HG-fermented groups. Moreover, the HG fermented by *B**. subtilis* KU43, *B**. subtilis* KU201, and *B**. polyfermenticus* SCD showed significantly lower mRNA levels of *TRP-1* and *TRP-2* in the MSH-stimulated cells.

## 4. Discussion

An imbalance between the generation of ROS and the radical-scavenging activity could lead to oxidative damage to macromolecules, which is connected to the acceleration of inflammation and the generation of melanin [9,23,25]. However, antioxidant compounds, which act as radical-scavenging agents, can attenuate the ROS in the cellular environment. Ginseng exerts radical-scavenging effects owing to its diverse phytochemical compounds that mediate ROS generation [23]. In a previous study, the radical-scavenging activity was significantly higher in HG, which possesses higher amounts of phenolic compounds and ginsenosides than normal soil-cultured ginseng [8]. The antioxidative effect was elevated after the fermentation of HG with *Leuconostoc mesenteroides* KCCM 12010P, which enzymatically converted the compounds to low-molecular-weight phenolics [15]. Moreover, a ginseng extract fermented with *Bacillus subtilis* KFRI 1127 demonstrated the highest contents of total sugar, phenolics, and acidic polysaccharides, as well as the highest antioxidant activity, compared with those of *Lactobacillus-* and *Pediococcus-*fermented extracts [17]. Similar to previous studies, the TPC, TFC, and free-radical-scavenging activity (ABTS and FRAP) also increased when the HG was subjected to fermentation in this study. Although it is difficult to identify those specific constituents of HG that possess these complicated properties and multiple functionalities, the increase in the phenolic compound concentration and flavonoid content followed by fermentation may influence the enhancement of the radical scavenging activity, as assessed by the ABTS and FRAP assays [3].

Macrophages are one of the main cells that regulate the inflammatory reaction by producing various inflammatory mediators, such as ROS, NO, PGE_2_, and cytokines [26]. Increased levels of NO and PGE_2_ formation are modulated by iNOS and COX-2 through the nuclear factor kappa B (NF-κB) pathway [19]. LPS, triggering the transduction of the inflammatory signaling system through transmembrane receptor 4 (TLR4), causes the activation of the NF-κB and mitogen-activated protein kinase (MAPK) pathways, and produces inflammation-related cytokines, such as TNF-α, IL-1β, and IL-6 [13]. Therefore, it is important to alleviate these key inflammatory factors in order to block the initiation of inflammatory responses. In this study, the fermented HG showed decreased amounts of NO and PGE_2_, accompanied by the inhibited mRNA expressions of cytokines (*TNF-α, IL-1β, and IL-6*). *Bifidobacterium longum* promoted the anti-inflammatory and antinociceptive activities of ginseng extract by aiding the production of ginsenosides Rg3, compound K, and Rh2 during fermentation [27]. Although, in most studies, ginsenosides, such as Rb1, Rg1, Rg3, Rh2, and compound K, inhibited the expressions of inflammatory cytokines (TNF-α, IL-1β, and IL-6) and enzymes (iNOS and COX-2), the inflammation-inhibiting properties of ginseng extracts were confirmed to be derived from a wide range of diverse components [28]. Ginseng oligopeptides improved the inflammatory response system by regulating the MAPK and NF-κB pathways induced by dextran in rat models [4]. Likewise, *Saccharomyces cerevisiae* GIW-1 fermentation was found to promote the acidic polysaccharide and uronic acid levels, which was attributed to the higher hydroxyl and superoxide anion radical-scavenging activity and a lower serum aspartate transaminase level and inflammation-related cytokines [29].

Melanocytes, which are located in the epidermal tissue, deliver melanin to adjacent keratinocytes, resulting in the organization of the pigmented skin layer [9]. Two varieties of melanin (pheomelanin and eumelanin) are generated from tyrosine through a tyrosinase-dependent pathway [30]. In the predominantly reported cAMP-dependent PKA–CREB pathway in melanogenesis, ultraviolet radiation triggers MITF activation, thereby regulating tyrosinase, TRP-1, and TRP-2 [31]. Tyrosinase plays a key role in pigment formation, which commences the modification of tyrosine via hydroxylation to L-DOPA, which is oxidated to L-dopaquinone. L-dopachrome, modified from L-dopaquinone, is isomerized to 5,6-dihydroxyindole-2-carboxylic acid (DHICA) with TRP-2 activation. Moreover, TRP-1 finally leads to melanogenesis by oxidizing DHICA to indole-5,6-quinone-2-carboxylic acid [32]. Thus, blocking the tyrosinase-related signaling pathways prevents the early stage of melanogenesis and assists in the skin-whitening reaction. According to Hwang et al. [33], quercetin and esculetin, which comprise most of the phenolic compounds in ginseng leaves, have been reported to inhibit pigmentation accumulation. In addition, the ethyl acetate extract of ginseng exhibits superior antioxidant activity and suppresses melanin generation in α-MSH-induced B16 cells [34]. Fermented HG gradually decreases the tyrosinase activity and melanogenesis in B16F10 cells by suppressing the tyrosinase-dependent pathway. These results strongly suggest that fermented HG is more potent at inhibiting melanogenesis and decreasing the melanogenic factors than nonfermented HG.

Consequently, it was demonstrated that the antioxidant, anti-inflammatory, and antimelanogenic effects of the fermented HG in the experimental material were particularly enhanced compared with those of the nonfermented HG. Because HG fermentation using *Bacillus* strains to produce high therapeutic functionality can be implemented industrially, it is possible to utilize fermented HG in various ways, including as a multifunctional raw material for formulating healthy functional foods and cosmetics.

## 5. Conclusions

In conclusion, this study aimed to identify the antioxidant, anti-inflammatory, and antimelanogenic effects of HG fermented with *Bacillus* strains. Because enzymes from microbial strains drive the bioconversion of phytochemicals, the HG fermented by *Bacillus subtilis* KU43, *Bacillus subtilis* KU201, *Bacillus polyfermenticus* SCD, and *Bacillus polyfermenticus* KU3 showed a dramatic increase in the phenolic and flavonoid levels, leading to promoted antioxidant activity. The cellular NO formation, PGE_2_ concentration, and expressions of inflammatory cytokines were reduced in the cells treated with the fermented HG, and especially *B*. *subtilis* KU201. In addition, the fermented HG blocked melanogenesis by inhibiting the *MITF, tyrosinase, TRP-1*, and *TRP-2* expressions. To sum up, it appears that the biofunctional activities of HG, such as the antioxidant, anti-inflammatory, and antimelanogenic effects, are improved through fermentation by *Bacillus* strains. Therefore, bacterial-fermented HC could potentially be used to develop functional food or cosmetic ingredients.

## Figures and Tables

**Figure 1 antioxidants-11-01848-f001:**
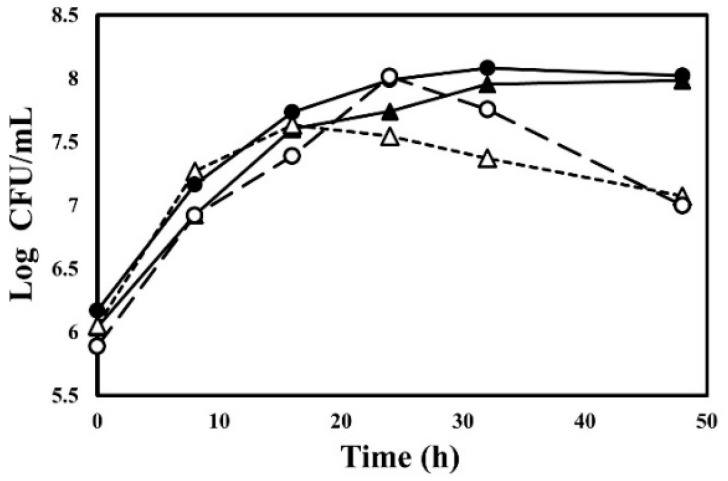
Change in bacterial growth over 48 h in hydroponic ginseng (HG): **△**: *Bacillus subtilis* KU43; **●**: *B. subtilis* KU201; **▲**: *B. polyfermenticus* SCD; **○**: *B. polyfermenticus* KU3.

**Figure 2 antioxidants-11-01848-f002:**
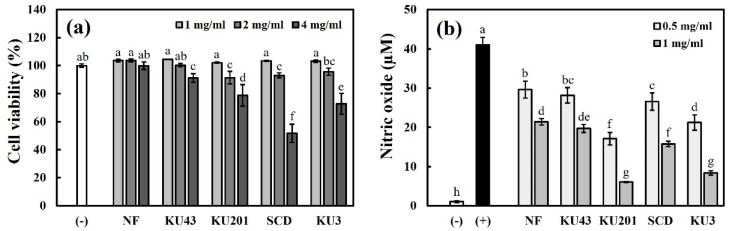
Cell viability and nitric oxide (NO) production in RAW 264.7 cells: (**a**) cell viability; (**b**) production of cellular NO. (−): negative control without lipopolysaccharide (LPS); (+): positive control with LPS; NF: nonfermented hydroponic ginseng (HG); KU43: HG fermented with *B. subtilis* KU43; KU201: HG fermented with *B. subtilis* KU201; SCD: HG fermented with *B. polyfermenticus* SCD; KU3: HG fermented with *B. polyfermenticus* KU3. The data are represented as means ± standard deviations of triplicates. Statistical differences (*p* < 0.05) are represented with different letters above error bars.

**Figure 3 antioxidants-11-01848-f003:**
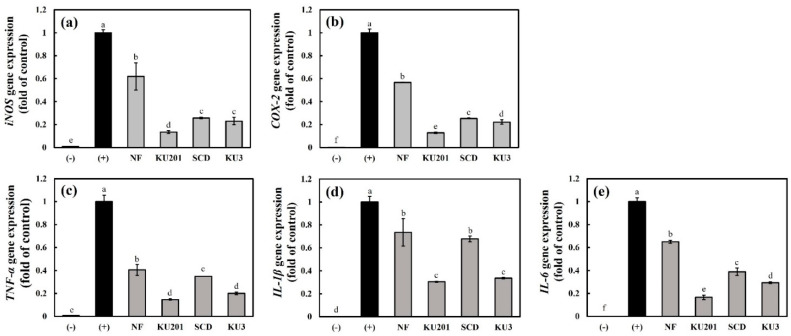
Relative gene expression levels of inflammation mediators in RAW 264.7 cells: (**a**) *inducible nitric oxide synthase (iNOS)*; (**b**) *cyclooxygenase-2 (COX-2)*; (**c**) *tumor necrosis factor (TNF)-α*; (**d**) *interleukin (IL)-1β*; (**e**) interleukin *IL-6*. (−): negative control without lipopolysaccharide (LPS); (+): positive control with LPS; NF: nonfermented hydroponic ginseng (HG); KU201: HG fermented with *B. subtilis* KU201; SCD: HG fermented with *B. polyfermenticus* SCD; KU3: HG fermented with *B. polyfermenticus* KU3. The data are represented as means ± standard deviations of triplicates. Statistical differences (*p* < 0.05) are represented with different letters above error bars.

**Figure 4 antioxidants-11-01848-f004:**
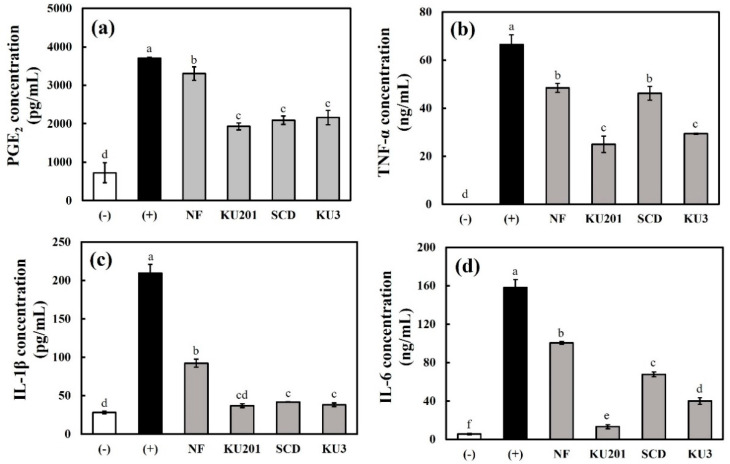
Protein concentrations of inflammatory cytokines in RAW 264.7 cells: (**a**) prostaglandin E_2_ (PGE_2_); (**b**) tumor necrosis factor (TNF)-α; (**c**) interleukin (IL)-1β.; (**d**) interleukin IL-6. (−): negative control without lipopolysaccharide (LPS); (+): positive control with LPS; NF: nonfermented hydroponic ginseng (HG); KU201: HG fermented with *B. subtilis* KU201; SCD: HG fermented with *B. polyfermenticus* SCD; KU3: HG fermented with *B. polyfermenticus* KU3. The data are represented as means ± standard deviations of triplicates. Statistical differences (*p* < 0.05) are represented with different letters above error bars.

**Figure 5 antioxidants-11-01848-f005:**
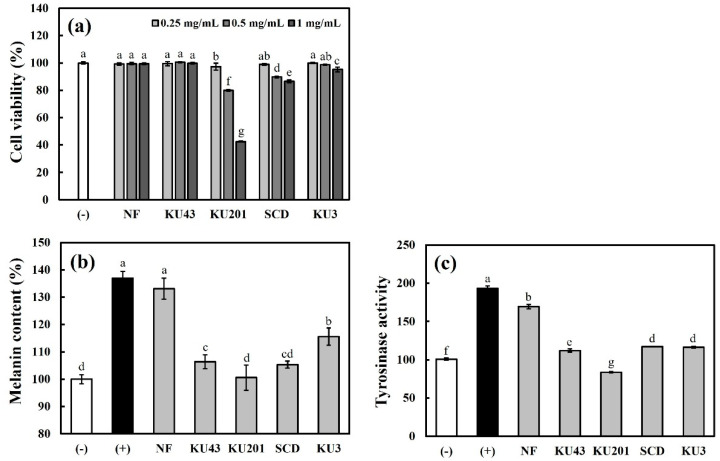
Cell viability, cellular melanin content, and cellular tyrosinase activity in B16F10 cells: (**a**) cell viability; (**b**) melanin content; (**c**) tyrosinase activity. (−): negative control without α-melanocyte-stimulating hormone (α-MSH); (+): positive control with α-MSH; NF: nonfermented hydroponic ginseng (HG); KU43: HG fermented with *B. subtilis* KU43; KU201: HG fermented with *B. subtilis* KU201; SCD: HG fermented with *B. polyfermenticus* SCD; KU3: HG fermented with *B. polyfermenticus* KU3. The data are represented as means ± standard deviations of triplicates. Statistical differences (*p* < 0.05) are represented with different letters above error bars.

**Figure 6 antioxidants-11-01848-f006:**
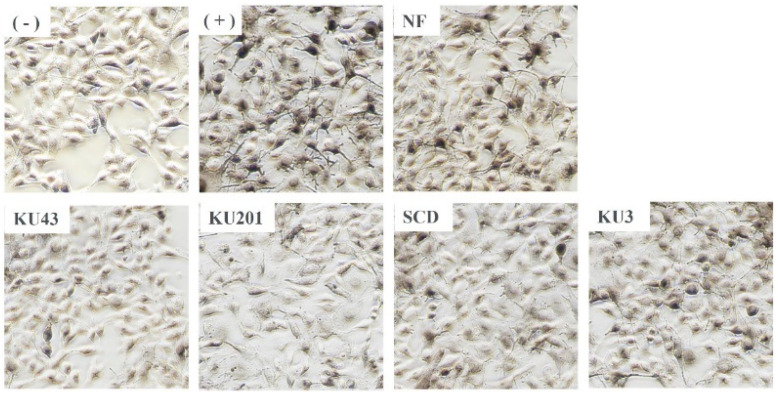
L-3,4-dihydroxyphenylalanine (L-DOPA) staining in B16F10 cells. (−): negative control without α-melanocyte-stimulating hormone (α-MSH); (+): positive control with α-MSH; NF: nonfermented hydroponic ginseng (HG); KU43: HG fermented with *B. subtilis* KU43; KU201: HG fermented with *B. subtilis* KU201; SCD: HG fermented with *B. polyfermenticus* SCD; KU3: HG fermented with *B. polyfermenticus* KU3.

**Figure 7 antioxidants-11-01848-f007:**
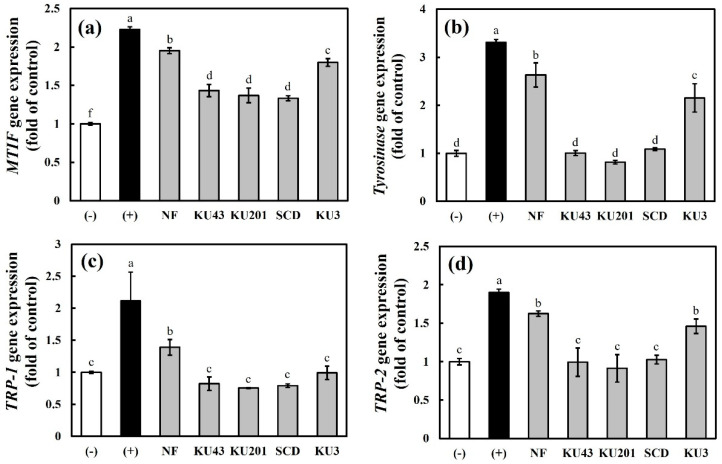
Relative gene expression levels of melanogenesis mediators in B16F10 cells: (**a**) *microphthalmia-associated transcription factor (MITF)*; (**b**) *tyrosinase*; (**c**) *tyrosinase-related protein (TRP)-1*; (**d**) *TRP-2*. (−): negative control without α-melanocyte-stimulating hormone (α-MSH); (+): positive control with α-MSH; NF: nonfermented hydroponic ginseng (HG); KU43: HG fermented with *B. subtilis* KU43; KU201: HG fermented with *B. subtilis* KU201; SCD: HG fermented with *B. polyfermenticus* SCD; KU3: HG fermented with *B. polyfermenticus* KU3. The data are represented as means ± standard deviations of triplicates. Statistical differences (*p* < 0.05) are represented with different letters above error bars.

**Table 1 antioxidants-11-01848-t001:** Primer sequences for detection of mRNAs.

Gene ^1^	Forward	Reverse	Accession Number
RAW 264.7 cells			
*iNOS*	CCCTTCCGAAGTTTCTGGCAGCAGC	GGCTGTCAGAGCCTCG-TGGCTTTGG	NM_010927
*C* *OX-2*	CACTACATCCTGACCCACTT	ATGCTCCTGCTTGAGTATGT	NM_011198
*TNF-α*	TTGACCTCAGCGCTGAGTTG	CCTGTAGCCCACGTCGTAGC	NM_013693
*IL-1β*	CAGGATGAGGACATGAGCACC	CTCTGCAGACTCAAACTCCAC	NM_008361
*IL-6*	GTACTCCAGAAGACCAGAGG	TGCTGGTGACAACCACGGCC	NM_031168
*β-Actin*	GTGGGCCGCCCTAGGCACCAG	GGAGGAAGAGGATGCGGCAGT	NM_007393
B16F10 cells			
*MITF*	TTACCAACAACCTCGGCACCAT	CTCCTGGCGACACTGATGACA	NM_001113198
*Tyrosinase*	CCTCCTGGCAGATCATTTGT	GGTTTTGGCTTTGTCATGGT	NM_011661
*T* *RP* *-1*	TTGCTGTAGTGGCTGCGTTGTT	AGGAGAGGCTGGTTGGCTTCAT	NM_031202
*T* *RP* *-2*	GCAAGAGATACACGGAGGAAG	CTAAGGCATCATCATCATCACTAC	NM_010024
*β-A* *ctin*	AGCCATGTACGTAGCCATCC	CTCTCAGCTGTGGTGGTGAA	NM_007393

^1^*iNOS*: *inducible nitric oxide synthase*; *COX-2*: *cyclooxygenase-2*; *TNF-α*: *tumor necrosis factor-α*; *IL-1β*: *interleukin-1β*; *IL-6*: *interleukin-6*; *β-Actin*: *reference gene*; *MITF*: *microphthalmia-associated transcription factor*; *TRP-1*: *tyrosinase-related protein-1*; *TRP-2*: *tyrosinase-related protein-2*.

**Table 2 antioxidants-11-01848-t002:** Total phenolic content (TPC), total flavonoid content (TFC), and antioxidant activities of nonfermented and fermented hydroponic ginseng (HG).

Samples	Nonfermented HG	Fermented HG			
	NF	*B. subtilis* KU43	*B. subtilis* KU201	*B. polyfermeticus* SCD	*B. polyfermeticus* KU3
TPC (mg GAE/g)	51.62 ± 1.50 ^e^	66.34 ± 2.47 ^d^	90.22 ± 1.50 ^a^	76.48 ± 0.57 ^b^	71.57 ± 0.57 ^c^
TFC (mg QE/g)	24.33 ± 0.71 ^c^	36.27 ± 1.75 ^b^	43.27 ± 4.99 ^a^	42.04 ± 2.85 ^a^	26.39 ± 2.14 ^c^
ABTS ^1^ (%)	25.30 ± 0.01 ^e^	34.00 ± 0.01 ^d^	51.34 ± 0.01 ^a^	36.88 ± 0.01 ^c^	40.62 ± 0.01 ^b^
FRAP ^1^ (μM)	132.10 ± 2.19 ^d^	176.30 ± 0.00 ^c^	236.27 ± 11.42 ^a^	207.86 ± 9.34 ^b^	199.66 ± 1.09 ^b^

^1^ ABTS: azino-bis(3-ethylbenz-thiazoline-6-sulfonic acid (ABTS); FRAP: ferric reducing antioxidant power. The data are represented as means ± standard deviations of triplicates. Statistical differences are represented with different letters (*p* < 0.05).

## Data Availability

Data are contained within the article.

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
