# Peer review of "Improved Antioxidative, Anti-Inflammatory, and Antimelanogenic Effects of Fermented Hydroponic Ginseng with Bacillus Strains"

_antioxidants, 2022, doi:10.3390/antiox11101848_

Round 1

Reviewer 1 Report

The manuscript is examining the improved antioxidative, anti-inflammatory, and anti-melanogenic effects of fermented hydroponic ginseng with Bacillus Strains. The manuscript is well designed, and the results are very interesting. 

There are a few comments from the reviewer as bellow;

Do authors have any reason to use a beta-actin primers with a different sequence in Raw 264.7 and B16F10 cells?

For the expression of mice and rats, gene (mRNA) symbols should be italicized, with only the first letter in upper-case (e.g., Il-6). Protein symbols are not italicized, and all letters are in upper-case. Thus, author should check the expression of genes in the manuscript.

Author Response

  1. Do authors have any reason to use a beta-actin primers with a different sequence in Raw 264.7 and B16F10 cells?
  • Thank you for your valuable comment. The different sequences of beta-actin were used in RAW 264.7 and B16F10 cells, respectively, according to each reference as follows. The different sequences in this study were used due to a different cell line.
    • Wang, H.M.; Qu, L.Q.; Ng, J.P.L.; Zeng, W.; Yu, L.; Song, L.L.; Wong, V.K.W.; Xia, C.L.; Law, B.Y.K. Natural Citrus Flavanone 5-Demethylnobiletin Stimulates Melanogenesis through the Activation of CAMP/CREB Pathway in B16F10 Cells. Phytomedicine 2022, 98, 153941.
    • Kim, H.S.; Yu, H.-S.; Lee, J.H.; Lee, G.W.; Choi, S.J.; Chang, P.-S.; Paik, H.-D. Application of Stabilizer Improves Stability of Nanosuspended Branched-Chain Amino Acids and Anti-Inflammatory Effect in LPS-Induced RAW 264.7 Cells. Food Sci. Biotechnol. 2018, 27, 451–459. /CREB Pathway in B16F10 Cells. Phytomedicine 2022, 98, 153941.
  1. For the expression of mice and rats, gene (mRNA) symbols should be italicized, with only the first letter in upper-case (e.g., Il-6). Protein symbols are not italicized, and all letters are in upper-case. Thus, author should check the expression of genes in the manuscript.
  • Thank you for your detailed comment. Expressions of gene and protein were checked and corrected. All of gene (mRNA) symbols were italicized, but abbreviation of gene symbols were written in upper-case as generally expression. Please refer the references as follows.
    • Song, M. W.; Park, J. Y.; Lee, H. S.; Kim, K. T.; Paik, H. D. Co-Fermentation by Lactobacillus brevis B7 Improves the Antioxidant and Immunomodulatory Activities of Hydroponic Ginseng-Fortified Yogurt. Antioxidants 2021, 10, 1447.
    • Gómez-Pastor, R.; Garre, E.; Pérez-Torrado, R.; Matallana, E. Trx2p-dependent regulation of Saccharomyces cerevisiae oxidative stress response by the Skn7p transcription factor under respiring conditions. Plos one 2013, 8, e85404.
    • Wissing, T. B.; Bonito, V.; van Haaften, E. E.; van Doeselaar, M.; Brugmans, M. M.; Janssen, H. M.; Smits, A. I. Macrophage-driven biomaterial degradation depends on scaffold microarchitecture. Frontiers in bioengineering and biotechnology 2019, 7, 87.

Author Response

Page 2, line 92, please indicate how much HG powder was refluxed with 50% ethanol during preparation of HG and how the yield is obtained.

  • Thank you for your valuable comment. The process of ginseng extract that 50 g of HG powder was refluxed with 500 mL of 50% ethanol at 65 °C for 6 h is added in Line 91-93.

Page 6, line 243, since only two levels of HG were tested, it is not exactly in a dose-dependent manner.

  • Thank you for your kind comment. Since it is tested with only two level of HG, I deleted a term ‘dose-dependent manner’ as you mentioned (Line 243).

Reviewer 3 Report

This manuscript present relevant information on the functional properties of phenolic compounds from fermented hydroponic ginseng. The study was well conducted and is well written. However, slight modifications should be done publication in this prestigious Journal.

Lines 10 – 12: This point should be discussed in deep into the introduction section

Line 104: It has been reported that the Folin-Ciocalteu assay is not a specific assay for determining phenolic compounds due to the presence of nonphenolic compounds, having an aromatic ring within the molecule can also reduce the reagent. Then, an explanation should be given to justify the use of this method and include this justification in the text.

Author Response

  1. Lines 10 – 12: This point should be discussed in deep into the introduction section
  • Thank you for your valuable comment. As you said, I deleted this part in `Abstract’ because there is no content about this in the `Introduction’. Also, I revised `Abstract’ part (Line 10, 11) based on the introduction (Line 38-44).
  1. Line 104: It has been reported that the Folin-Ciocalteu assay is not a specific assay for determining phenolic compounds due to the presence of nonphenolic compounds, having an aromatic ring within the molecule can also reduce the reagent. Then, an explanation should be given to justify the use of this method and include this justification in the text.
  • Thank you for your detailed comment. Although the Folin-Ciocalteu assay is not a specific assay for determining phenolic compounds, this assay has been used generally for phenolic compound by many researchers (1, 2). Therefore, the content of phenolic compounds in hydroponic ginseng could be compared with prior studies which used this same method (Line 104, 212).
    • Hwang, J.E.; Suh, D.H.; Kim, K.-T.; Paik, H.-D. Comparative Study on Anti-Oxidative and Anti-Inflammatory Properties of Hydroponic Ginseng and Soil-Cultured Ginseng. Food Sci. Biotechnol. 2019, 28, 215–224.
    • Hwang, J.E.; Kim, K.-T.; Paik, H.-D. Improved Antioxidant, Anti-Inflammatory, and Anti-Adipogenic Properties of Hydroponic Ginseng Fermented by Leuconostoc Mesenteroides KCCM 12010P. Molecules 2019, 24, 3359.